# Decision-Making in Implantology—A Cross-Sectional Vignette-Based Study to Determine Clinical Treatment Routines for the Edentulous Atrophic Mandible

**DOI:** 10.3390/ijerph18041596

**Published:** 2021-02-08

**Authors:** Michael Korsch, Winfried Walther, Bernt-Peter Robra, Aynur Sahin, Matthias Hannig, Andreas Bartols

**Affiliations:** 1Dental Academy for Continuing Professional Development, Karlsruhe, Lorenzstrasse 7, 76135 Karlsruhe, Germany; winfried_walther@za-karlsruhe.de (W.W.); andreas_bartols@za-karlsruhe.de (A.B.); 2Clinic of Operative Dentistry, Periodontology and Preventive Dentistry, University Hospital, Saarland University, Building 73, 66421 Homburg, Germany; matthias.hannig@uks.eu; 3Private Practice, Center for Implantology and Oral Surgery, Berliner Str. 41, 69120 Heidelberg, Germany; 4Institute of Social Medicine and Health Services Research, Otto-von-Guericke-University of Magdeburg, Leipziger Str. 44, 39120 Magdeburg, Germany; bp.robra@gmail.com; 5Private Practice, Blumenstrasse 5, 69115 Heidelberg, Germany; aynursahin75@gmx.de; 6School for Dental Medicine, Christian-Albrechts-University Kiel, Clinic for Conservative Dentistry and Periodontology, Arnold-Heller-Straße 3, 24105 Kiel, Germany

**Keywords:** dental implant, specialists, oral surgeon, maxillofacial surgeon, pre-implantological treatment, bone augmentation, bone resection, edentulous mandible

## Abstract

This cross-sectional study aimed to investigate the influence of possible factors in the patient history on decision making in the therapy for a severely atrophied edentulous mandible. A vignette-based survey among 250 maxillofacial and oral surgeons was conducted. Determinants that could influence the therapy decision were patient age, smoking, fear of surgery, and radiotherapy in the head and neck area (the implant region is not in the direct radiation area). To achieve a suitable implant site, the options offered to the surgeons were bone split, bone block, augmentation with bone substitute material, and bone resection. There also was the option of rejecting any therapy. The response rate was 47%. Patient age, radiotherapy, and fear of surgery did not influence the approval of a therapy. Smoking was associated with a significantly lower endorsement of a treatment. Resection was preferred by a large majority to all other forms of therapy, regardless of the four determinants. Surgeons tend to refrain from bone block transplants in older patients. In summary, it can be said that, of the four determinants, only smoking influenced treatment refusal. Bone resection is the preferred therapy independent of all determinants.

## 1. Introduction

Oral health has a major impact on general illnesses [1,2]. Conversely, numerous studies have proven the effects of general diseases on oral health [3,4]. However, little is known about the influence of certain patient factors on the treatment decisions of surgeons during oral surgery.

An adequate implant bed is a prerequisite for optimal implant positioning. After tooth loss, resorption of the alveolar bone occurs in all cases, which can vary in severity. Treatment methods, such as mini-invasive tooth extraction and the use of tension-free sutures, can reduce the resorption [5]. If the resorption of the alveolar ridge is advanced, the conditions for implant surgery become less favorable [6]. In cases with advanced alveolar ridge resorption, augmentation techniques can be used for bone reconstruction to enable successful implant-supported rehabilitation [7]. The pre-implantological treatment methods to achieve an adequate implant bed include additive (e.g., bone block and augmentation with bone substitute material) [7,8,9] and expansive (bone split) [10] techniques. Alternatively, resection of narrow bone crests can lead to a sufficient implant bed.

Augmentation and implantation can be accompanied by complications that can affect the surgical procedure and the prognosis. In general, augmentations have a higher risk of complications than the implantological procedure itself [11,12]. Every additional intervention increases the risk of surgery. In addition to the surgical skills and experience of the surgeon, risk factors and possibly patient interest in implants or fear of surgical interventions might also influence the decision to use a particular surgical technique.

With regard to risk factors, a distinction is made between exogenous, systemic, local, and iatrogenic risks. For example, radiotherapy as an exogenous risk factor has a negative influence on the bony blood flow and can lead to osteoradionecrosis [13]. Irradiation of the jaws, especially the surgical area, therefore has an increased risk of complications. Special measures, such as perioperative antibiotics, might be necessary in such cases. Other local factors are inadequate oral hygiene, periodontitis, and smoking [14,15]. Smoking leads to a reduced blood supply to the soft tissues [16]. This can have a negative influence on wound healing [17] and the incidence of inflammation could be significantly increased [18].

The patient’s interest in implantation, but also the fear of the surgical procedure, might also influence the surgeon in selecting a suitable therapeutic procedure. The avoidance of very invasive therapeutic procedures is likely to be determined primarily by the patient’s age and fear of surgery. In older patients, for example, it can be sensible to avoid more complex treatment methods with higher risks, longer treatment times, and increased surgical trauma and to prefer more minimally invasive techniques with less effort and shorter treatment times instead. One-stage operations are preferred, in which tissue-level implants are often used [19]. It is also known that dental fear leads patients to reject implantation [20]. Therefore, a careful medical history should be taken from the patient and taken into account in pre-implantological planning.

A method to investigate the influence of the above-described risk factors at different levels on the selection of augmentation methods by clinical specialists is case vignettes. Such case vignettes are typified case descriptions and are used in comparative health care research for the experimental investigation of health care decisions [21,22]. In terms of methodology, they are between theory and real patient cases. A prospective study has shown that case vignettes have comparable informative value in terms of medical practice as a systematic evaluation of medical records and standardized medical patients [23]. With the help of clinical case vignettes, the understanding of care processes can be deepened. They enable the scientific monitoring of innovative clinical developments [24]. Case vignettes provide a valid and comprehensive method that focuses directly on the treatment approach in clinical practice. They are a cost-effective method for measuring the quality of care provided by a group of physicians [23,25,26,27].

The present study aimed to investigate with help of case vignettes to what extent the factors “age”, “patient’s interest in implants” vs. “patient’s fear of implantation”, “co-morbidity”, and “smoking” influence the treatment decision of designated specialists in the therapy for the severely atrophied edentulous mandible. So far there is no knowledge of what influences the decision-making of the implantologists.

## 2. Materials and Methods

The present cross-sectional study was conducted as a vignette-based survey. For this purpose, a questionnaire was prepared and sent in hardcopy by mail to all maxillofacial surgeons as well as oral surgeons in the southern German states of Baden-Württemberg, Bavaria, Hesse, Rhineland-Palatinate, and Saarland. Together with the State Dental Chambers of the respective federal states, a total of 250 maxillofacial surgeons and oral surgeons working in private practice who were authorized to conduct specialist training in oral surgery could be reached. The existence of authorization for specialist training ensured that these surgeons are experienced specialists. All specialists to be surveyed were contacted by telephone before the questionnaire was sent out. In the telephone conversation, they were informed about the nature and aim of the survey. Consent to participate in the study was requested. Refusals were respected. If, after several attempts, contact was not possible, the interviewee received the questionnaire without special notice.

The study was conducted following the Declaration of Helsinki and the Professional Code for Physicians of the local Medical Council. The Ethics Committee of the Saarland Medical Council reviewed and approved the proposed study (Ref. No.: 133/11). The declaration of consent was obtained orally bythe participants and was confirmed by returning the questionnaire. Information was also provided about the positive ethical vote on this survey. Participants who did not want to take part in the study orally refused the declaration of consent or did not return the questionnaire.

The study followed the STROBE (Strengthening the Reporting of Observational studies in Epidemiology) guidelines for cross-sectional studies, where applicable (https://www.strobe-statement.org).

The following physician-related characteristics were collected: specialist for maxillofacial surgery or specialist for oral surgery; number of professional years after specialist training; postgraduate continuing education: completion of a Master of Science (MSc) or implantology curriculum.

### 2.1. Structure of the Questionnaire

The questionnaire, relevant for this publication, consisted of clinical case vignettes with the request to choose the preferred treatment option. The original questionnaire can be found in the Appendix A. This questionnaire contained two clinical case vignettes of the severely atrophied edentulous mandible. In both case vignettes, the bony situation could be assessed using X-rays. On the basis of these X-rays, the respondents were able to decide whether to opt for an additive, expansive, or resective technique for treatment to achieve an adequate implant bed. In order to investigate the influence of possible factors in the patient history on decision-making in the therapy, the anamneses varied randomly within the individual case vignettes.

### 2.2. Clinical Case Vignettes

The questionnaire contained case vignettes relating to the implantological therapy of the edentulous mandible to collect the indication routines of the respondents. The respondents were presented with two patient cases with the corresponding real clinical findings. The case vignettes contained an anamnesis, clinical findings, and X-rays (orthopantomograms and cone-beam computed tomography (CBCT) excerpts).

### 2.3. Variable Descriptors

Each case vignette had variable descriptors in the anamnesis. The two risk factors “smoking behaviour” as a local risk factor and the comorbidity “radiotherapy” as an external risk factor were chosen as one set of descriptors. Another set of descriptors were “age of the patient” and “anxiety”. In the individual vignettes, a low or high patient age, smoker or non-smoker, no or existing co-morbidity, and no or existing fear of surgery were specified. By using a random generator when creating the vignette sets, these vignette characteristics were distributed independently of each other. Utilizing such planned variance of individual determinants, decision patterns and causes for favoring a certain therapy can be determined. The given treatment options should cover the three methods mentioned above (additive, expansive, and resective methods) for creating a wide implant bed in a narrow atrophic alveolar ridge and should be common techniques used in surgical practice [28,29]. That is why the decision on the four treatment options “bone split”, “bone block”, “bone substitute material”, and “bone resection” was made.

Five maxillofacial and oral surgeons tested the questionnaire for practicability (comprehensibility, consistency of the content of the findings and measures) before sending it out to the intended sample.

### 2.4. Description of the Clinical Case Vignettes

Case Vignette 1 (“Lower jaw with residual dentition that cannot be preserved”):

This clinical case vignette was intended to assess the influence of age and smoking habit on the clinical decision.

The presented patient had an insufficient mandibular denture. The wish of the patient was a prosthetic restoration on implants. The pantomographic and CBCT findings (see Figure 1 and Figure 2) showed a reduced width of the mandible. The referring dentist aimed for a fixed prosthesis in the mandible on four implants. Teeth 33 and 43 were classified as not worth preserving. Thus, edentulism in the lower jaw loomed. The first vignette variance was the patient age (55 or 75 years), the second was the smoking status.

The following variants were therefore incorporated into the vignettes:

Combination 1: Age of the patient 55 years, patient is non-smoker.

Combination 2: Age of the patient 55 years, patient is a smoker.

Combination 3: Age of the patient 75 years, patient is non-smoker.

Combination 4: Age of the patient 75 years, patient is a smoker.

Case Vignette 2 (“toothless lower jaw”):

This clinical case vignette was intended to determine the influence of previous radiotherapy treatment in the cervical area and the patient’s attitude towards the intended treatment. The case involved a 73-year-old female patient who had been edentulous for ten years and could not cope with her complete denture in the lower jaw (repeated decubitus treatments). During the radiotherapy, the jawbone had not been in the radiation field and the salivary flow seemed to be unaffected. X-rays (orthopantomogram and CBCT, Figure 3 and Figure 4) in this case showed an atrophic lower and upper jaw with a narrow atrophic mandibular ridge. The referring doctor’s preference was a removable denture in the mandible on four implants.

The following variants were incorporated into the vignettes:

Combination 1: No systemic diseases, the patient is very interested in a prosthetic restoration on implants.

Combination 2: No systemic diseases, the patient is rather anxious and repeatedly demands an explanation of the procedure.

Combination 3: Radiation treatment in the neck area, the patient is very interested in a prosthetic restoration on implants.

Combination 4: Radiation treatment in the neck area, the patient is rather anxious and repeatedly demands an explanation of the procedure.

The surgeons surveyed were asked to indicate for both vignettes how they would proceed in each case. For this purpose, five uniform possible options were given: bone split, bone block, augmentation with a bone substitute material, resection, and general approval of the therapy.

For all therapy options, it was possible to tick “yes”, “not at all”, and “possibly”. The option “Yes” represented the therapy of the surgeon’s first choice, “by no means” that this therapy is not recommended to the opinion of the surgeon, and “possibly” stood for “I am considering this option, I will decide intraoperatively”.

The datasets used and/or analyzed during the current study are available from the corresponding author on reasonable request.

### 2.5. Data Evaluation

The data from the questionnaires was collected using Microsoft Excel and analyzed using IBM SPSS Statistics 21 (IBM SPSS Statistics, IBM, Armonk, NY, United States) on Windows XP.

The evaluation was performed with complete data sets. Missing data from the study participants were excluded on a case-by-case basis. Crosstabs were used as statistical procedures. Binary logistic regressions were used for the dependent characteristics. It was examined whether the respondent’s specialist designation (maxillofacial vs. oral surgeon) was associated with the preferred care. A probability of error of *p* < 0.05 was interpreted as a statistically significant finding.

## 3. Results

Out of the 250 questionnaires, 117 were returned. The response rate was thus 46.8%. The combinations of characteristics in the returned 117 questionnaires were distributed in a similar way to the sample sent out (about a quarter of 250 each) (Table 1).

### 3.1. Physician-Related Characteristics

Of the 117 participants who answered the questionnaire, 49 (42%) were specialists in maxillofacial surgery and 68 (58%) were specialists in oral surgery. Twenty-six (53%) of the 49 specialists in maxillofacial surgery had the additional designation of oral surgery. Thirteen (11%) of the 117 study participants were female. The average professional experience of the participants after completion of specialist training was 18.7 (SD 7.5) years. Sixty-four percent of all surgeons had completed a curriculum for implantology and 13% of the respondents had the additional title Master of Science (MSc). All those interviewed were experts in their field. Gender, work experience, completion of a curriculum for implantology, or Master of Science did not affect the results.

### 3.2. Outcome Scale for Case Vignette 1 (Edentulous Lower Jaw)

The statistical analysis showed (Figure 5) that 81% of the respondents generally were in favor of treatment and only 6% were ultimately against treatment, and 13% would “possibly” treat. Almost one in two (49%) indicated bone resection as the therapy of choice. A total of 25% of all surgeons favored bone replacement materials. Bone block grafts were favored by only 4%, but rejected by 64% of the respondents. Bone split as the therapy of choice was only indicated by 10%, but 43% indicated that they would not use this technique in this case.

The binary logistic regression did not reveal any association of the possible therapy modalities with the specialist designation (maxillofacial vs. oral surgeon), here parameterized as a dependent variable (Table 2).

The patient’s age did not influence the approval or rejection of therapy. However, the analysis showed a significant association between patient age (2 groups) and “bone block” and “bone substitute material” as a choice of therapy for edentulous jaws (Table 3). “Bone block” was more likely to be rejected in older patients and “bone substitute material” was more likely to be approved than in younger patients.

Smoking was not associated with the different possible choices of therapy, but smoking was associated with a significantly lower endorsement of treatment (Table 4). Nevertheless, the rejection of treatment in smokers was less than 10% (Figure 5).

Under “other”, the most frequent statements were “no fixed prosthesis on four implants” (*n* = 15, 13%) and “more implants for fixed restoration” (*n* = 5, 4%). Combinations, such as a bone resection together with bone substitute material or autologous bone mixed with bone substitute material, were also suggested. Plasma-rich growth factor (PRGF), the shell technique, the use of Astra Profile implants, and a smoking ban were also among the free texts.

### 3.3. Outcome Scale for Case Vignette 2 (Edentulous Lower Jaw)

The evaluation showed (Figure 6) that 82% of the respondents supported a therapy in Case Vignette 2 and only 2% refused treatment. More than 2/3 (68%) of the surgeons indicated resection as the therapy of choice. Bone block transplants were favored by only 12% but rejected by 76% of the respondents. Bone split as the treatment of choice was only proposed by 14%, but 61% indicated that this technique should not be used in this case.

The specialist designation (maxillofacial vs. oral surgeon) was not associated with any kind of therapy decision in the binary logistic regression analysis (Table 5). Radiation of the neck area did not lead to an intervention rejection. Radiotherapy was only significantly associated with the bone split technique (Table 6), where the rejection of a bone split was significantly higher. The fear of a surgical intervention had no statistically significant association with the possible therapy (Table 7).

Under “other”, the most frequent response was the use of diameter-reduced implants (*n* = 6, 5%) and only two instead of four implants (*n* = 3, 3%). Vestibuloplasty, PRGF, certain shell techniques, and the combination of bone resection with bone substitute material were also mentioned.

## 4. Discussion

Of the four determinants—patient age, smoking, radiotherapy, and fear of surgery—only smoking led significantly more often to the rejection of an intervention. Resection was preferred by a large majority to all other forms of therapy, regardless of the four determinants. So far there is little knowledge of what influences the decision-making of implantologists and to what extent certain pre-implantological techniques are favored.

One of the study’s strengths was the vignette-based survey method used here, which enabled a large number of respondents. The response rate of almost 47% achieved can be classified as high, especially against the background that no incentives were provided in this cross-sectional study. Mehlkop and Becker stated in their study that a response rate of approximately 28% was to be expected if the respondents did not receive any reward, and approximately [30] 52% if an incentive was provided. Nevertheless, one of the weaknesses of the present study is that 53% of the surgeons did not answer. Their expertise would certainly have had an impact on the results. Another weak point relates to determinants. The term “smoking” is not clearly defined. A smoker can have one or 40 cigarettes a day. The influence on the organism and certainly on the therapy decision can be very different. A general formulation was also chosen for the determinant “fear of surgery“. Fear can be very complex. No radiation dose was given for radiotherapy. Different radiation doses certainly have different effects on the surrounding tissue and possibly also on the therapy decision. The survey technique of the vignette, however, made it possible to determine and also to compare the influence of different determinants under the same clinical conditions (atrophied edentulous mandible) on the therapy decision.

Case-based vignettes have a comparable validity as a systematic medical record evaluation or an evaluation with standardized patients [23,25,26,27]. This method is used in medicine, but is not very widespread in dentistry and has not yet been used in dental implantology [31,32,33]. With Case Vignette 1, it could be shown that more than 80% of the surgeons approved of therapy for the narrow atrophic edentulous mandible. While the patient’s age did not influence the therapy decision, smoking led to increased rejection of therapy. However, patient age was significantly associated with the bone block and bone substitute material. In older patients, surgeons tended to refrain from complex procedures and avoid bone donor regions. Concerning bone block transplants, it can be assumed that patients of higher age are more likely to suffer from postoperative discomfort and should be spared a longer treatment period. The higher level of refusal of treatment among smokers can be explained by the fact that tobacco consumption leads to higher infection rates and can promote peri-implantitis in the long term [34,35]. The comparison of the determinants “age” and “smoking” showed that age had a lower influence on the decision to recommend a therapy than smoking. The participants, therefore, decided not to do “age-rationing”, but to take into account the oral health status, which experience has shown to be associated with smoking. Reasons for this could be that placing implants in the lower jaw is successful in the long term even at an advanced age [36] and higher patient satisfaction can be achieved [37].

In Case Vignette 2, a therapy was somewhat more advocated. Fear of surgery and radiotherapy did not influence the approval of a therapy. Patients’ anxiety did not correlate with the treatment methods chosen. Another scientific publication also showed that care planning is influenced by the patient’s dental needs, not by their phobic status [33]. An already performed radiotherapy treatment in the neck area was only significantly correlated with a bone split. This option was only rarely chosen. The most serious complication of radiotherapy is radiation necrosis of the bone [38]. This is an irreversible, progressive devitalization of the irradiated bone. The clinical manifestation may be pain, orofacial fistulas, exposed necrotic bone, pathological fractures, and putrid infections [39]. For these reasons, radiotherapy is considered a risk factor [40]. The fact that many surgeons were nevertheless in favor of surgery may be because the mandible was not in the direct radiation field. Therefore, no statement can be made on how the therapy decision would have turned out if the implant site had been in the direct radiation field.

The specialist designation (maxillofacial vs. oral surgeon) was not associated with any kind of therapy decision in both vignettes. Bone resection was preferred by a large majority over any other treatment option. In all four combinations of the two case vignettes, bone resection was the “therapy of my choice”. Although resection was favored by surgeons in both case vignettes, there are numerous other treatment options. These depend on the severity of the atrophy, the patient’s medical history, the surgeon’s surgical knowledge, and many other factors. Therefore, it is not yet possible to choose an evidence-based treatment method for the edentulous mandible [41,42,43].

While bone resection usually can be done with simultaneous placement of implants and avoids augmentation procedures, bone block transplantation often requires a second operation a few months later [29,44,45]. Therefore, bone resection is particularly suitable for elderly and anxious patients and avoids a complex procedure in post-radiation conditions. The use of bone substitute material followed at a considerable distance. This seems to be reasonable since both techniques usually allow simultaneous placement of implants and the surgical risk seems to be acceptable. Bone block and bone split are the more complex therapy forms of the four options. Furthermore, block transplantation with autologous bone always requires a donor region, which leads to additional trauma with chances of increased postoperative symptoms and increased surgical risk [46]. Bone block transplants and bone splits were rejected with high percentages and had only a few supporters. This could be explained by the reasons mentioned above. In a review by de Groot et al. in 2018, it was shown that the survival rate of dental implants in the edentulous mandible is high regardless of the bone augmentation method used [47]. While an extraoral donor region for the ridge reconstruction can possibly be avoided with the two case vignettes used here, this is often not possible in the case of massive atrophy [48].

The present cross-sectional vignette-based study showed that, in the atrophic mandible, bone resection is by far the preferred therapy by maxillofacial and oral surgeons for patients of all ages, smoking behavior, previous radiotherapy treatment, and fear of surgery. The results showed that the survey participants take the abovementioned patient characteristics into account when making therapy decisions. Comparative studies should be carried out on the superiority of individual treatment methods over others.

## 5. Conclusions

In summary, it can be said that, of the four determinants, only smoking influenced the rejection or approval of the treatment. Bone resection is the preferred therapy independent of all determinants.

The study results show that different factors have a different influence on the surgeon’s decision-making. No factor leads to a uniform decision for or against treatment for the surgeons. Surgeons assess the influence of these factors differently. Nevertheless, only four of the many possible influencing factors for the therapy decision could be discussed in this study. 

## Figures and Tables

**Figure 1 ijerph-18-01596-f001:**
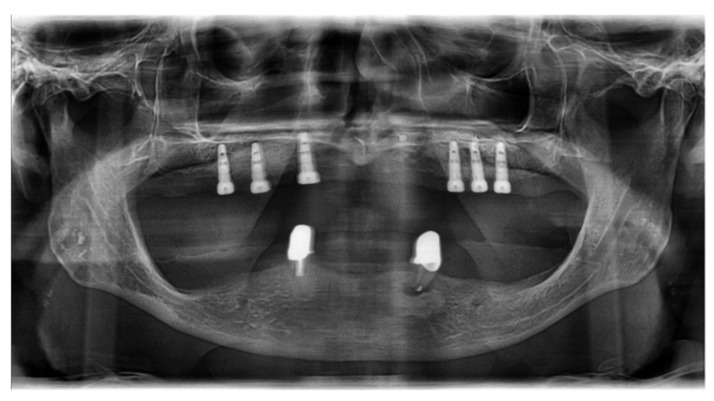
Panoramic tomography of the first case vignette. In the lower jaw, the remaining teeth 33 and 43 are present. Tooth 33 shows apical periodontitis. There is a bony height deficit in the premolar and molar areas of the lower jaw.

**Figure 2 ijerph-18-01596-f002:**
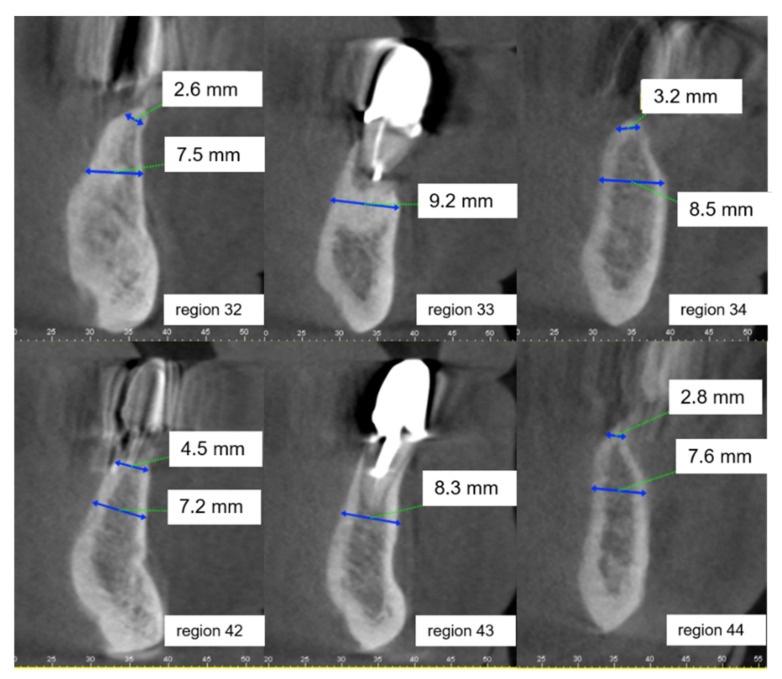
Sagittal plane from the CBCT of regions 32–34 and 42–44 of Case Vignette 1. The alveolar ridge is substantially atrophied in the crestal region. The widths of the alveolar ridge (mm = millimeters) are provided in the respective cut-outs. The alveolar ridge height is sufficient.

**Figure 3 ijerph-18-01596-f003:**
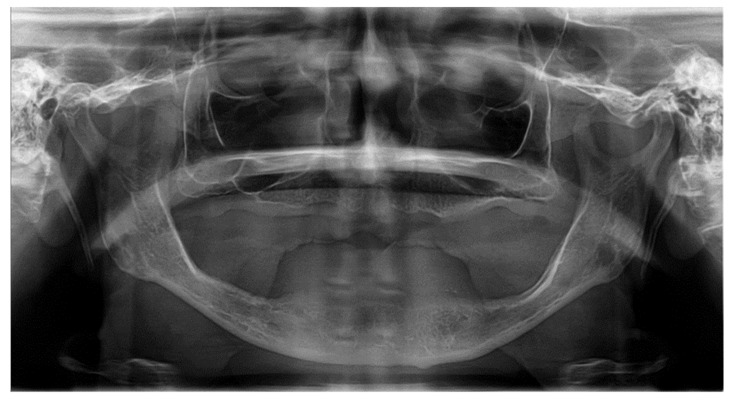
Panoramic radiograph of Case Vignette 2: The alveolar ridge is atrophied in both the upper and lower jaw. In the premolar and molar regions of the mandible, the bone height is insufficient. There is sufficient bone height interforaminally.

**Figure 4 ijerph-18-01596-f004:**
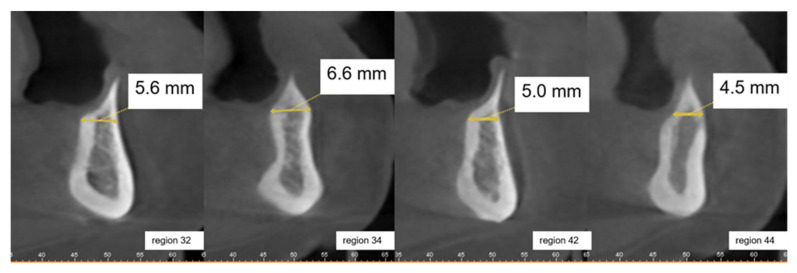
Sagittal planes from CBCT of regions 32, 34, 42, and 44 of Case Vignette 2. The alveolar ridge is substantially atrophied in the crestal region. The widths of the alveolar ridge (mm = millimeters) are provided in the respective cut-outs. The alveolar ridge height is sufficient.

**Figure 5 ijerph-18-01596-f005:**
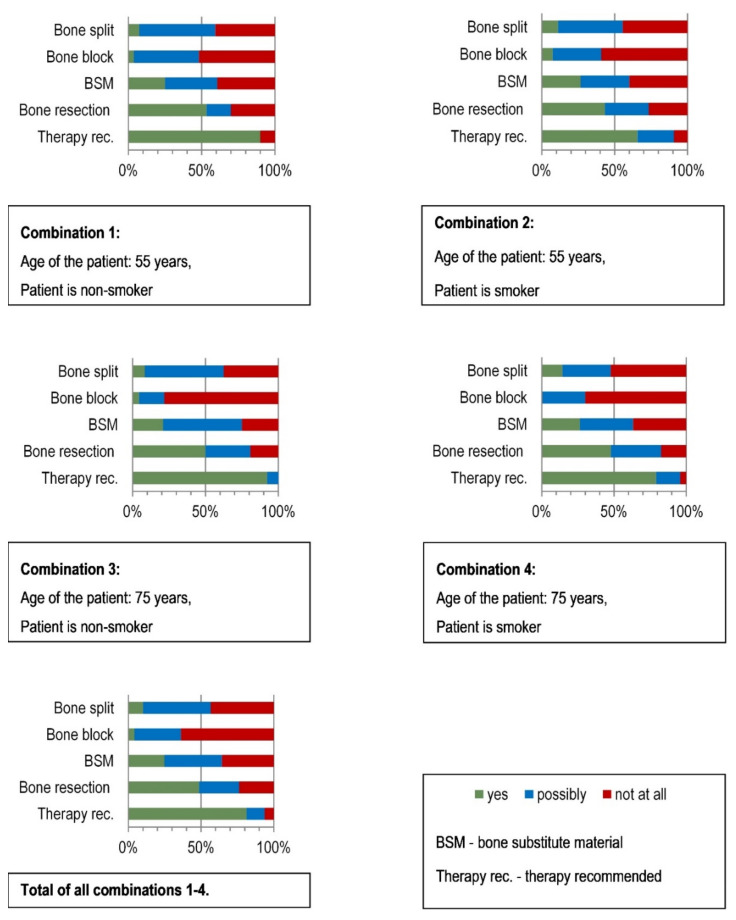
Preferred therapy decisions of the surgeons surveyed for the restoration of the edentulous severely atrophied mandible according to the four different combinations of “patient age” and “smoking”. Case Vignette 1.

**Figure 6 ijerph-18-01596-f006:**
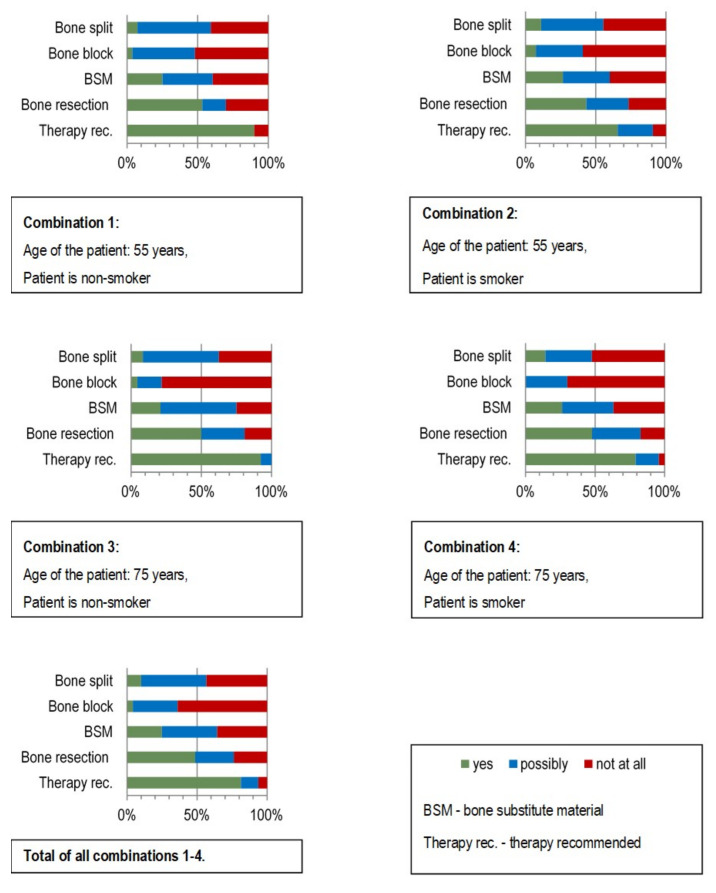
Preferred therapy decisions of the surgeons surveyed for the restoration of the edentulous severely atrophied mandible according to the four different combinations of “radiotherapy” and “anxiety”. Case Vignette 2.

**Table 1 ijerph-18-01596-t001:** Distribution of the vignette combinations of the returned questionnaires (*n* = 117).

	Case Vignette 1	Case Vignette 2
	*n* (%)	*n* (%)
Combination 1	31 (27)	29 (25)
Combination 2	35 (30)	28 (24)
Combination 3	26 (22)	35 (30)
Combination 4	25 (21)	25 (21)

**Table 2 ijerph-18-01596-t002:** Association between the dependent variable “specialist designation” (maxillofacial vs. oral surgeon) and the independent variables “bone split”, “bone block”, “bone substitute material”, “bone resection”, and “therapy recommended” for Case Vignette 1.

	Regression Coefficient B	Standard Error	Forest	df	*p*-Value	Exp(B)
Bone split	0.189	0.249	0.575	1	0.448	1.208
Bone block	0.069	0.253	0.075	1	0.784	1.072
Bone substitute material	−0.262	0.266	0.973	1	0.324	0.769
Bone resection	0.207	0.315	0.431	1	0.511	1.230
Therapy recommended	0.263	0.302	0.758	1	0.384	1.301
Constant	−0.675	0.467	2.090	1	0.148	0.509

Nagelkerke R-squared = 0.50.

**Table 3 ijerph-18-01596-t003:** Association between the dependent variable “patient age” (55 years vs. 75 years) and the independent variables “bone split”, “bone block”, “bone substitute material”, “bone resection”, and “therapy recommended” for Case Vignette 1.

	Regression Coefficient B	Standard Error	Forest	df	*p*-Value	Exp(B)
Bone split	0.137	0.257	0.284	1	0.594	1.146
Bone block	0.532	0.268	3.951	1	0.047	1.702
Bone substitute material	−0.556	0.280	3.953	1	0.047	0.574
Bone resection	−0.337	0.323	1.091	1	0.296	0.714
Therapy recommended	0.217	0.316	0.473	1	0.491	1.243
Constant	0.533	0.469	1.294	1	0.255	1.704

Nagelkerke R-squared = 0.50.

**Table 4 ijerph-18-01596-t004:** Association between the dependent variable “smoking” (smoker vs. non-smoker) and the independent variables “bone split”, “bone block”, “bone substitute material”, “bone resection”, and “therapy recommended” for Case Vignette 1.

	Regression Coefficient B	Standard Error	Forest	df	*p*-Value	Exp(B)
Bone split	0.323	0.265	1.482	1	0.223	1.381
Bone block	−0.182	0.271	0.452	1	0.501	0.833
Bone substitute material	0.226	0.277	0.666	1	0.415	1.254
Bone resection	−0.369	0.333	1.229	1	0.268	0.691
Therapy recommended	−1.080	0.374	8.356	1	0.004	0.340
Constant	0.508	0.482	1.111	1	0.292	1.662

Nagelkerke R-squared = 0.50.

**Table 5 ijerph-18-01596-t005:** Association between the dependent variable “specialist designation” (maxillofacial vs. oral surgeon) and the independent variables “bone split”, “bone block”, “bone substitute material”, “bone resection”, and “therapy recommended” for Case Vignette 2.

	Regression Coefficient B	Standard Error	Forest	df	*p*-Value	Exp(B)
Bone split	0.175	0.268	0.424	1	0.515	1.191
Bone block	−0.438	0.380	1.332	1	0.248	0.645
Bone substitute material	−0.303	0.257	1.396	1	0.237	0.738
Bone resection	0.777	0.416	3.495	1	0.062	2.175
Therapy recommended	0.175	0.282	0.387	1	0.534	1.191
Constant	−0.987	0.574	2.960	1	0.085	0.373

Nagelkerke R-squared = 0.50.

**Table 6 ijerph-18-01596-t006:** Association between the dependent variable “previous radiotherapy” (yes vs. no) and the independent variables “bone split”, “bone block”, “bone substitute material”, “bone resection”, and “therapy recommended” for Case Vignette 2.

	Regression Coefficient B	Standard Error	Forest	df	*p*-Value	Exp(B)
Bone split	0.607	0.276	4.829	1	0.028	1.835
Bone block	0.611	0.354	2.989	1	0.084	1.843
Bone substitute material	−0.423	0.262	2.610	1	0.106	0.655
Bone resection	0.135	0.384	0.124	1	0.724	1.145
Therapy recommended	−0.330	0.296	1.246	1	0.264	0.719
Constant	−0.075	0.546	0.019	1	0.890	0.927

**Table 7 ijerph-18-01596-t007:** Association between the dependent variable “readiness to involvement” (anxiousness vs. interest in implant therapy) and the independent variables “bone split”, “bone block”, “bone substitute material”, “bone resection”, and “therapy recommended” for Case Vignette 2.

	Regression Coefficient B	Standard Error	Forest	df	*p*-Value	Exp(B)
Bone split	−0.231	0.257	0.805	1	0.369	0.794
Bone block	0.386	0.334	1.337	1	0.248	1.472
Bone substitute material	−0.307	0.247	1.544	1	0.214	0.735
Bone resection	0.153	0.371	0.171	1	0.679	1.166
Therapy recommended	−0.117	0.279	0.176	1	0.675	0.890
Constant	0.393	0.534	0.540	1	0.462	1.481

Nagelkerke R-squared = 0.50.

## Data Availability

Data is contained within the article or Appendix A.

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
