# Peer review of "Decision-Making in Implantology—A Cross-Sectional Vignette-Based Study to Determine Clinical Treatment Routines for the Edentulous Atrophic Mandible"

_ijerph, 2021, doi:10.3390/ijerph18041596_

Round 1

Reviewer 1 Report

Dear authors I appreciate your theory and the entire manuscript. 
I think it is very well conducted. I asked for few adjustments.

Pag. 2, line 58-60, you are talking about the possibility to reduce the bone resorption by several pre-implantological methods. I suggest you to cite this interesting article that hypotize a new consept about bone resorption after tooth extraction that may be prevented by a mini-invasive surgery.

I'd re-write this sentence:

"This goal can be achieved with various pre-implantological treatment methods."

in this way:

"This goal can be achieved with various pre-implantological treatment methods such as a mini-invasive tooth extraction and the use of tension free sutures."

Adding this citation:

Marconcini S, Denaro M, Cosola S, Gabriele M, Toti P, Mijiritsky E, et al. Myofibroblast Gene Expression Profile after Tooth Extraction in the Rabbit. Materials (Basel). 2019 Nov 9;12(22):3697. doi: 10.3390/ma12223697.

Again in the introduction i would re-write this sentence:

" In older patients, for example, it can be sensible to avoid 82
more complex treatment methods with higher risks, longer treatment times, and increased surgical trauma and to prefer more minimally invasive techniques with less effort and shorter treatment times instead. "

talking about also the type of minimally invasive surgey: two-stage, one-stage, bone level implants of tissue level citing this review on the topic because the tissue level implant with one stage surgery could be usefull in patients with high risk of complications.

Cosola S, Marconcini S, Boccuzzi M, Menchini Fabris GB, Covani U, Peñarrocha-Diago M, et al. Radiological Outcomes of Bone-Level and Tissue-Level Dental Implants: Systematic Review. Int J Environ Res Public Health. 2020 Sep 22;17(18):6920. doi: 10.3390/ijerph17186920. 

For the rest, I really like your article and I hope this is accepted.

Author Response

Reviewer 1

Dear authors I appreciate your theory and the entire manuscript.

I think it is very well conducted. I asked for few adjustments.

  1. Pag. 2, line 58-60, you are talking about the possibility to reduce

the bone resorption by several pre-implantological methods. I

suggest you to cite this interesting article that hypotize a new

consept about bone resorption after tooth extraction that may be

prevented by a mini-invasive surgery.

I'd re-write this sentence:

"This goal can be achieved with various pre-implantological

treatment methods."

in this way:

"This goal can be achieved with various pre-implantological

treatment methods such as a mini-invasive tooth extraction and the

use of tension free sutures."

Adding this citation:

Marconcini S, Denaro M, Cosola S, Gabriele M, Toti P, Mijiritsky E,

et al. Myofibroblast Gene Expression Profile after Tooth Extraction

in the Rabbit. Materials (Basel). 2019 Nov 9;12(22):3697. doi:

10.3390/ma12223697 .

  • Obviously, we have not made our statement clear on this point. We actually wanted to say that in cases in which resorption has already occurred, pre-implantological treatment methods like augmentation or resection must be carried out in order to achieve an adequate implant bed. We have made this point more understandable.
  • Nevertheless, we consider the comment made by the reviewer to be good and have included the following information:

„After tooth loss, resorption of the alveolar bone occurs in all cases, which can vary in severity. Treatment methods such as mini-invasive tooth extraction and the use of tension-free sutures can reduce the resorption.”

  1. Again in the introduction i would re-write this sentence:

" In older patients, for example, it can be sensible to avoid 82

more complex treatment methods with higher risks, longer

treatment times, and increased surgical trauma and to prefer more

minimally invasive techniques with less effort and shorter treatment

times instead. "

talking about also the type of minimally invasive surgey: two-stage,

one-stage, bone level implants of tissue level citing this review on

the topic because the tissue level implant with one stage surgery

could be usefull in patients with high risk of complications.

© 1996-2021 MDPI (Basel, Switzerland) unless otherwise stated Disclaimer Terms and Conditions

(https://www.mdpi.com/about/terms-and-conditions)

Privacy Policy (https://www.mdpi.com/about/privacy)

Cosola S, Marconcini S, Boccuzzi M, Menchini Fabris GB, Covani

U, Peñarrocha-Diago M, et al. Radiological Outcomes of BoneLevel and Tissue-Level Dental Implants: Systematic Review. Int J

Environ Res Public Health. 2020 Sep 22;17(18):6920. doi:

10.3390/ijerph17186920 .

  • We added the following:

“One-stage operations are preferred, in which tissue level implants are often used.“

For the rest, I really like your article and I hope this is accepted.

Reviewer 2 Report

This is a very good paper regarding the “Decision-making in implantology – A vignette-based study to determine clinical treatment routines for the edentulous atrophic mandible”. This study aimed to investigate the influence of possible factors in the patient history on decision making in the therapy for severely atrophied edentulous mandible, throughout a vignette-based survey among 250 maxillofacial and oral surgeons. The authors, however, need to address some points in order to improve the manuscript. I highlighted some critical issues:

  1. The authors need to include the study design. As this study was carried out through the distribution of a questionnaire, it is a cross-sectional study that should be mentioned in the text in the material and methods section and then in the discussion.
  2. There are important guidelines for all types of clinical studies, in this case as it is a cross-sectional study the STROBE (Srengthening the Reporting of Observational studies in Epidemiology) guidelines should be mentioned and the STROBE check list followed.
  3. The discussion section should be improved. The discussion section basically presented the results of the study and almost not discussed with the literature on the subject. Systematic reviews should be included in the discussion and compared with the results obtained in this study.

Author Response

Reviewer 2

This is a very good paper regarding the “Decision-making in

implantology – A vignette-based study to determine clinical

treatment routines for the edentulous atrophic mandible”. This

study aimed to investigate the influence of possible factors in the

patient history on decision making in the therapy for severely

atrophied edentulous mandible, throughout a vignette-based

survey among 250 maxillofacial and oral surgeons. The authors,

however, need to address some points in order to improve the

manuscript. I highlighted some critical issues:

  1. The authors need to include the study design. As this study

was carried out through the distribution of a questionnaire, it is

a cross-sectional study that should be mentioned in the text in

the material and methods section and then in the discussion.

  • We mentioned cross-sectional study in the title, abstract, methods and discussion.

  1. There are important guidelines for all types of clinical studies, in

this case as it is a cross-sectional study the STROBE

(Srengthening the Reporting of Observational studies in

Epidemiology) guidelines should be mentioned and the

STROBE check list followed.

  • We followed the STROBE guidelines in the publication and added the STROBE list as an appendix.

  1. The discussion section should be improved. The discussion

section basically presented the results of the study and almost

not discussed with the literature on the subject. Systematic

reviews should be included in the discussion and compared

with the results obtained in this study.

  • The present study is a novelty in dental implantology. Reviews on the subject of therapy decisions in implantology are therefore not available. However, further reviews on the individual treatment methods were included in the discussion.

Reviewer 3 Report

The research is interesting but I have some suggestion:

In my opinion, to have a more objective answer was better not to put in one vignette case two possibilities of the two different anamneses. It was better to create a separate case with one anamnesis.

The material and method is to poor. The Authors should put more information in “2.1. Structure of the questionnaire” in it than “ The questionnaire, relevant for this publication, consisted of clinical case vignettes with the request to choose the preferred treatment option. The original questionnaire can be found in the supplemental files”

We cannot find why the Authors suggested this type of treatments in their case( is not supported this information even in the introduction)

We have no information that can be very interesting, for example, how long experience has the surgeons. Maybe much more on the decision of the treatment have the experience.? It could be useful to have more information about the participants.

Author Response

Reviewer 3

The research is interesting but I have some suggestion:

  1. In my opinion, to have a more objective answer was better not to

put in one vignette case two possibilities of the two different

anamneses. It was better to create a separate case with one

anamnesis.

  • In our study we were interested in the effects of the determinants on the therapy decision. A factorial permutation of these was therefore carried out within a vignette/anamnesis in order to identify effects. Different anamneses within a vignette are therefore necessary.

  1. The material and method is to poor. The Authors should put more

information in “2.1. Structure of the questionnaire” in it than “ The

questionnaire, relevant for this publication, consisted of clinical

case vignettes with the request to choose the preferred treatment

option. The original questionnaire can be found in the

supplemental files”

  • The following has been changed or added:

“This questionnaire contained two clinical case vignettes of the severely atrophied edentulous mandible. In both case vignettes, the bony situation could be assessed using x-rays. On the basis of these x-rays, the respondents were able to decide whether to opt for an additive, expansive, or resective techniques for treatment to achieve an adequate implant bed. In order to investigate the influence of possible factors in the patient history on decision making in the therapy, the anamneses varied randomly within the individual case vignettes.”

  1. We cannot find why the Authors suggested this type of treatments

in their case( is not supported this information even in the

introduction)

  • The following has been changed or added:

“The given treatment options should cover the three methods mentioned above (additive, expansive and resective methods) for creating a wide implant bed in a narrow atrophic alveolar ridge and should be common techniques used in surgical practice. That is why the decision on the four treatment options bone split”, “bone block”, “bone substitute material” and “bone resection” was made.”

  1. We have no information that can be very interesting, for example,

how long experience has the surgeons. Maybe much more on the

decision of the treatment have the experience.? It could be useful

to have more information about the participants.

  • The following has been changed or added:

Methods

The present study was conducted as a vignette-based survey. For this purpose, a questionnaire was prepared and sent in hardcopy by mail to all maxillofacial surgeons as well as oral surgeons in the southern German states of Baden-Württemberg, Bavaria, Hesse, Rhineland-Palatinate and Saarland. Together with the State Dental Chambers of the respective federal states, a total of 250 maxillofacial surgeons and oral surgeons working in private practice who were authorized to conduct specialist training in oral surgery could be reached.

The following physician-related characteristics were collected: specialist for maxillofacial surgery or specialist for oral surgery; number of professional years after specialist training; postgraduate continuing education: Completion of a Master of Science (MSc) or implantology curriculum.

Results

Physician-related characteristics

Of the 117 participants who answered the questionnaire, 49 (41.9%) were specialists in maxillofacial surgery, and 68 (58.1%) were specialists in oral surgery. Twenty six (53.1%) of the 49 specialists in maxillofacial surgery had the additional designation of oral surgery. Thirteen (11%) of the 117 study participants were female. The average professional experience of the participants after completion of specialist training was 18.7 (SD 7.5) years. Sixty-four percent of all surgeons had completed a curriculum for implantology and 13% of the respondents had the additional title Master of Science (MSc). All interviewed were experts in their field. Gender, work experience, completion of a curriculum for implantology, or Master of Science did not affect the results.

Round 2

Reviewer 2 Report

Accept in present form

Reviewer 3 Report

Accept in the present form